# Plasma Proteomic Analysis in Morquio A Disease

**DOI:** 10.3390/ijms22116165

**Published:** 2021-06-07

**Authors:** José V. Álvarez, Susana B. Bravo, María Pilar Chantada-Vázquez, Sofía Barbosa-Gouveia, Cristóbal Colón, Olalla López-Suarez, Shunji Tomatsu, Francisco J. Otero-Espinar, María L. Couce

**Affiliations:** 1Department of Forensic Sciences, Pathology, Gynecology and Obstetrics, Pediatrics, Neonatology Service, Health Research Institute of Santiago de Compostela (IDIS), Hospital Clínico Universitario de Santiago de Compostela, CIBERER, MetabERN, 15706 Santiago de Compostela, Spain or josevictor.Alvarezgonzalez@nemours.org (J.V.Á.); cristobal.colon.mejeras@sergas.es (C.C.); Olalla.elena.lopez.suarez@sergas.es (O.L.-S.); 2Health Research Institute of Santiago de Compostela (IDIS), CIBERER, MetabERN, 15706 Santiago de Compostela, Spain; sofia.bsg@gmail.com; 3Skeletal Dysplasia Lab Nemours Biomedical Research Nemours/Alfred I, duPont Hospital for Children, 1600 Rockland Road, Wilmington, DE 19803, USA; shunji.tomatsu@nemours.org; 4Proteomic Platform, Health Research Institute of Santiago de Compostela (IDIS), Hospital ClínicoUniversitario de Santiago de Compostela, 15706 Santiago de Compostela, Spain; sbbravo@gmail.com (S.B.B.); Mariadelpilarchantadavazquez@gmail.com (M.P.C.-V.); 5Paraquasil Platform, Health Research Institute of Santiago de Compostela (IDIS), Hospital Clínico Universitario de Santiago de Compostela, 15706 Santiago de Compostela, Spain; 6Department of Pharmacology, Pharmacy and Pharmaceutical Technology, School of Pharmacy, Campus Vida, University of Santiago de Compostela, 15872 Santiago de Compostela, Spain

**Keywords:** biomarkers, enzyme replacement therapy, lysosomal disorders, proteomics

## Abstract

Mucopolysaccharidosis type IVA (MPS IVA) is a lysosomal disease caused by mutations in the gene encoding the enzyme*N*-acetylgalactosamine-6-sulfate sulfatase (GALNS), and is characterized by systemic skeletal dysplasia due to excessive storage of keratan sulfate (KS) and chondroitin-6-sulfate in chondrocytes. Although improvements in the activity of daily living and endurance tests have been achieved with enzyme replacement therapy (ERT) with recombinant human GALNS, recovery of bone lesions and bone growth in MPS IVA has not been demonstrated to date. Moreover, no correlation has been described between therapeutic efficacy and urine levels of KS, which accumulates in MPS IVA patients. The objective of this study was to assess the validity of potential biomarkers proposed by other authors and to identify new biomarkers. To identify candidate biomarkers of this disease, we analyzed plasma samples from healthy controls (*n*=6) and from untreated (*n*=8) and ERT-treated (*n*=5, sampled before and after treatment) MPS IVA patients using both qualitative and quantitative proteomics analyses. The qualitative proteomics approach analyzed the proteomic profile of the different study groups. In the quantitative analysis, we identified/quantified 215 proteins after comparing healthy control untreated, ERT-treated MPSIVA patients. We selected a group of proteins that were dysregulated in MPS IVA patients. We identified four potential protein biomarkers, all of which may influence bone and cartilage metabolism: fetuin-A, vitronectin, alpha-1antitrypsin, and clusterin. Further studies of cartilage and bone samples from MPS IVA patients will be required to verify the validity of these proteins as potential biomarkers of MPS IVA.

## 1. Introduction

Mucopolysaccharidosis type IVA (MPS IVA, OMIM #253000) or Morquio A syndrome is an autosomal recessive disease caused by mutations in the *GALNS* gene. This molecular defect results in a deficiency in the enzyme N-acetylgalactosamine-6-sulfate sulfatase (GALNS, E.C.3.1.6.4) [1,2,3], which degrades the specific glycosaminoglycans(GAGs) keratan sulfate (KS) and chondroitin-6-sulfate (C6S) [4,5,6,7]. These GAGs accumulate in multiple tissues, mainly bone, cartilage, heart valves, and cornea, leading to progressive systemic skeletal dysplasia [8,9]. Significant non-skeletal manifestations, including respiratory disease, spinal cord compression, cardiac disease, impaired vision, hearing loss, and dental problems, have also been described in MPS IVA patients [10,11,12,13]. MPS IVA is more frequently associated with severe and extensive skeletal manifestations than the other MPS types. Specifically, hypermobility of the joints is a characteristic of MPS IVA that distinguishes this disease from other forms of MPS. Furthermore, cognitive involvement is generally absent in MPS IVA [14]. Among untreated patients, respiratory failure is the primary cause of death, which typically occurs during the second or third decades of life [15,16].

Currently, two therapies for MPS IVA are used in clinical practice: enzyme replacement therapy (ERT) with recombinant human GALNS; and hematopoietic stem cell transplantation (HSCT) [5,11]. ERT and HSCT are based on the cross-correction principle, where by lysosomal enzymes are taken up by the cells of deficient recipients and targeted to the lysosomes via the mannose-6-phosphate receptor. However, as described for other lysosomal storage diseases, ERT for MPS IVA suffers from several limitations, including the need for weekly infusions of 4–6 h; rapid clearance due to a short half-life (35 min in human, 2 min in mouse) [17,18,19]; a high price [19,20]; limited penetration of the avascular cartilage; and immunological responses against the infused enzyme [16,17,21]. Moreover, clinical trials have shown little improvements in terms of bone growth and skeletal dysplasia. There is no definitive therapy that markedly improves bone and cartilage lesions in MPS IVA [14,18,19,20]. Potential biomarkers proposed in recent studies include KS, C6S, and blood levels of collagen II [22,23]. Although clinical trials have studied urinary KS as a potential biomarker, it has not been demonstrated that decreases in this parameter correlate with clinical improvement. Urinary KS originates in the kidneys but does not reflect the involvement of bone or other relevant tissues in MPS IVA, and is considered a pharmacokinetic marker but not a surrogate biomarker. The identification of valid biomarkers of bone lesions to evaluate therapeutic efficacy thus remains an unmet need. While bone cartilage samples are the best samples in which to search for candidate biomarkers of MPS IVA, the objective of this study was to assess the validity of candidate plasma biomarkers proposed in previous studies and to search for new biomarkers. To this end, we carried out both quantitative and qualitative proteomic analyses of plasma samples from.

Within rare disease research, proteomics is a fast-growing field. Approaches based on liquid chromatography-tandem mass spectrometry (LC-MS/MS), which are used to search for biomarkers in other human diseases [24,25,26,27,28], are key tools used to study normal and affected tissues and body fluids (e.g., plasma) in rare diseases [29].

Disease states involve alterations in the expression levels, phosphorylation states, and post-transcriptional modifications of proteins. Therefore, proteomics techniques have been applied to the study of normal and pathological conditions to search for biological information that could further our understanding of pathophysiological mechanisms and identify novel disease biomarkers [26,28,30]. Specific disease biomarkers identified using proteomics techniques could help improve diagnostic and prognostic precision and serve as key tools for the evaluation of treatment outcomes [24,25,26,27,28,29,30,31]. We recently characterized new proteomic biomarker candidates in leukocytes of MPS IVA patients [32]. However, in the present study we sought to identify disease biomarkers in body fluids that can be more easily sampled and feasibly incorporated into a clinical setting to increase diagnostic accuracy in patients with this disease. We focused on potential biomarkers in plasma, which we identified using two proteomics techniques: qualitative LC-MS/MS and a quantitative assay that compared protein expression between controls and untreated or ERT-treated MPS IVA patients.

## 2. Results

Plasma samples were acquired from 13 MPS IVA patients and 6 controls and classified as follows: healthy controls; untreated MPS IVA patients; and MPS IVA patients who had been receiving weekly ERT infusions for a mean of 7 years. From these ERT-treated patients, samples were acquired 24 h before (ERT-a) and 24 h after (ERT-b) one of their weekly infusions (Figure 1). The demographic features of the participating MPS IVA patients have been previously described [32].

Qualitative and quantitative proteomic analyses of plasma proteins were performed on all patients and control samples.

### 2.1. Qualitative Analysis of Proteins in Plasma Samples

In this study, we identified all proteins in plasma samples using data-dependent acquisition (DDA)-LC-MS/MS. Subsequent analysis was limited to proteins identified with 1% error (i.e., those with a false discovery rate (FDR) < 1). To obtain the most representative proteins per group, we selected proteins identified in at least n-1 samples per group (where n is the number of samples per group) (Appendix A). The Venn diagram in Figure 2 depicts the distribution of proteins in each of the 4 groups, and indicates those that were common to multiple groups.

Once we determined the proteins characteristic of each group, we performed a functional analysis using the FunRich program. This analysis allows us to characterize the behavior of the proteins identified in each group. Comparison of ERT-a and ERT-b samples revealed a favorable evolution (i.e., values obtained in ERT-a or ERT-b were close to thoseobtained for control samples) for certain proteins. Next, we selected groups of proteins in which alterations are observed in MPS IVA, and analyzed their behavior in each study group. From these, we identified proteins involved in hemostasis, bone regeneration, reorganization of theextracellular matrix and collagen, the immune response, complement system, and oxidative stress.

Among proteins implicated in hemostasis (blood coagulation, fibrin clot formation, and blood coagulation intrinsic pathway), we observed that values obtained for ERT-a samples (i.e., those collected 6 days after the preceding enzyme infusion) were closer to those of control samples than to ERT-b samples (Figure 3a). This result suggests that blood levels of these proteins require several days to stabilize after ERT infusion. A similar effect was observed for proteins implicated in different aspects of hemostatic processes, including platelet activation, platelet aggregation, and platelet degranulation (Figure 3b).

Other proteins identified in the proteomic analysis were implicated in bone regeneration and collagen function. Analysis of the behavior of these proteins showed that ERT had a negative regulatory effect on some (including proteins involved in mineralization, ossification, and osteoblast differentiation) and a positive regulatory effect on others (bone resorption and regulation of bone mineralization) (Figure 4a). In the case of proteins that were negatively regulated, ERT required more than 24 h to affect protein expression. Conversely, in the case of positively regulated proteins (involved in bone resorption and osteoblast differentiation) plasma levels increased after treatment in the ERT-b group (Figure 4a).

Proteomic analysis of proteins involved in collagen function revealed values close to those of the control group for proteins related to a positive regulation of the collagen biosynthesis (Figure 4b) in the ERT-a group. Moreover, in the ERT-a study group, we observed decreases in the expression of proteins implicated in collagen biosynthesis and collagen fibril organization. The behavior of proteins involved in metabolic and catabolic processes was unaltered after ERT administration (data not shown).

A search for proteins that interact with collagen in the extracellular matrix identified 5 proteins; P02671, P02751 (fibronectin 1), P02675, P51884 (lumican), and P02679.

We detected a relatively high number of proteins involved in the immune response. The results of the functional analysis of these proteins are shown in Figure 5 and Figure 6. Figure 5a shows the results of functional enrichment analysis of proteins implicated in inflammation, including proteins related to interleukins (ILs) IL-1, IL-1β, IL-6, IL-8, and to interferon γ (TNF-γ). Proteins were grouped according to their participation in different processes, including the cellular response to interferon γ, the cellular response to IL-1, the cellular response to IL-6, positive regulation of IL-1 secretion, positive regulation of IL-1β secretion, negative regulation of IL-1β secretion, negative regulation of IL-6 production, and positive regulation of IL-8 biosynthesis. In all cases, we observed a reduction in the expression of the proteins of interest24 h after ERT administration (ERT-b), except for proteins involved in positive regulation of IL-1 secretion (no change observed after ERT). In UG we detected 1.79% of proteins related to IL-1, IL-1β, and TNF-γ. This corresponding value in the ERT-a group (1.25%) was closer to that observed in controls (1.23%). In the case of proteins related to IL-6 and IL-8 we detected values of 0.88% in UG,0.63% in ERT-a, and 0.61 in the control group.

The results of the functional enrichment analysis of proteins involved in biological processes related tothe immune response, immunoglobulin production, positive regulation of B cell activation, negative regulation inflammatory response, and leukocyte migrationare shown in Figure 5b. In all cases, ERT increased the expression of proteins involved in each of these biological processes, except for negative regulation of the inflammatory response.

Functional enrichment analysis was also performed to identify protein partner expression between groups of proteins related to inflammatory processes (biological processes relating to (i) the complement system, including the classical complement activation pathway, (ii) general complement activation, and (iii) the alternative complement activation pathway). The results (Figure 6) reveal a positive effect of ERT, although levels of protein expression levels did not reach those observed in the control group (Figure 6a). Figure 6b shows the results of the functional enrichment analysis of proteins involved in the oxidative stress response. Treatment (ERT-a) resulted in protein levels similar to those observed in the control group for proteins involved in both complement activation and oxidative stress.

### 2.2. SWATH Quantitative Analysisof Proteins in Plasma Samples

Quantitative analysis of proteins using SWATH revealed significant alterations in the expression of multiple proteins in the ERT-a and ERT-b groups relative to the control group (Table 1).

#### Quantitative Analysis Using SWATH

Proteins that were upregulated or down regulated in each group may be relevant in MPS IVA disease. Differences in protein expression in the UG may reveal candidate biomarkers of disease. Proteins upregulated or downregulated in the ERT-a and ERT-b groups can be considered as markers of disease evolution as they were altered before treatment. We identified certain proteins that were downregulated by more than 100 fold, including AMBP, ASM3B, KV117, and ZBTB4 (highlighted in dark red in Figure 7 and Table 2).

Of the proteins that were the upregulated and downregulated in the UG, ERT-a, and ERT-b groups relative to controls, 31% were related to immune system function. It is important to point out that most of the proteins for which changes in expression were observed in the study groups were down regulated. In addition to upregulated protein in the UG, ERT-a, and ERT-b groups relative to CG, we alsodetected2 immunoglobulins (KV320 and LV321) that were not present in the CG.

Some of these downregulated proteins (highlighted in red in Table 2 and in the heatmap in Figure 8) may be of particular importance in MPS IVA.

In order to shown the results in a better way, we represent a heatmap using the proteins up and down regulated shown in Table 2. Moreover, the proteins were classified into different functions and were highline in different colors. Among these proteins, those shown in Figure 9 and Figure 10 stand out.

The SWATH analysis identified proteins that were downregulated in the UG and the ERT-a and b groups relative to controls, or were absent entirely. These included FA10 and SAA4, which were absent in all patient groups, and RET4 and C1QB, which were absent in UG and in UG and ERT-a, respectively (see Figure 10 and Figure 11).

In the UG group, we detected changes in expression in 4 proteins with respect to the 3 other groups (Figure 12 and Table 3).

Because these proteins were likely dysregulated as a consequence of the disease, they may include potential disease biomarkers that are either up-or down regulated in MPS IVA.

Theproteinsfor which changes in expression were detected in the UG included severalimmunoglobulins. This is unsurprising, given the important inflammatory component of MPS. Moreover, proteins that were dysregulated in this group included some previously proposed as biomarkers of MPS (e.g., A1AT[alpha 1 antitrypsin]) [33].We also observed dysregulation of proteins involved in inflammation (S10A9), binding to collagen (C1QC), and prothrombin (THRB), a thyroid hormone receptor that mediates the biological activities of thyroid hormone [34]. Proteins that were downregulated in the UG included C1QB (in line with previously reports, Figure 9) and the serine protease inhibitor A2AP (Figure 10 and Table 4).

The most important dysregulated protein identified in this analysis was A1AT. Analysis of A1AT levels in individuals in each of the different groups (Figure 13) revealed significant upregulation in the UG compared with each of the other groups. A1AT levels in the ERT-a and ERT-b groups were comparable to those observed in controls. This protein may constitute not only a biomarker of MPSIVA, but also an indicator of disease progression, potentially providing a better means of monitoring treatment responses in MPS IVA.

While treatment resulted in changes in the expression of proteins in both the ERT-a and ERT-b groups (Table 4), none of these proteins were common to both groups.

In this analysis, we identified proteins that were expressed in only one of the treated groups (ERT-a or ERT-b) and were absent from all other groups. SAA1 and APOF weredetected exclusively in the ERT-a group and TPIS, QSOX1, and KV108exclusively in the ERT-b group. Changes in the expression of these proteins can be considered a consequence of ERT. More detailed analyses will be required to determine their role in the treatment response. GELS was upregulated in the ERT-b group compared with controls (Figure 11). The fact that this protein was upregulated in the untreated group, and even more so in the ERT-b group, suggests that its expression is increased as a consequence of the disease process.

## 3. Discussion

Quality surrogate biomarkers to monitor disease progression and treatment response are currently lacking in MPS in general, and in Morquio A disease in particular. Currently, the diagnosis of MPS IVA is established by GAG assay, enzymatic activity analysis, and genetic studies. To date, the most useful biomarker for MPS IVA is GAG levels in urine and blood. High levels of KS and C6S are indicative of MPS IVA disease. However, decreases in urinary KS levels after ERT do not imply that treatment is working as urinary GAG originates from the kidneys and does not correlate with bone lesion severity in MPS IVA [4]. Blood KS could constitute a more appropriate readout of therapeutic effect, as it is directly derived from bone. Another problem is that GAG levels in MPS IVA patients decrease naturally with age and therefore may be normalized or nearly normalized without treatment due to destruction of the cartilage in which KS is predominantly synthesized [35]. While a recent study reported that the disaccharide chondroitin sulfate can be used to monitor treatment with ERT, to date this biomarker has only been studied in a small number of samples from MPS IVA patients. [36].

Proteomic studies can provide greater insight into the pathological mechanisms underlying diseases and help identify candidate biomarkers. Sample collection is complicated in MPS IVA, in which bone is the main tissue affected. Proteomic analyses of fibroblasts and leukocytes from MPS IVA patients published by our group [37] have demonstrated correction of the levels of certain dysregulated proteins following ERT. In the present study, we focused our analysis on the effects of MPS IVA and ERT on plasma proteins implicated in bone physiology, inflammatory processes, and oxidative stress. To this end, we performed both qualitative and quantitative analyses. Qualitative analyses revealed changes in the expression of proteins associated with coagulation, bone regeneration, collagen reorganization, inflammation, and oxidative stress. ERT reversed the dysregulation of plasma proteins involved in hemostasis and oxidative stress, but had no such effect on proteins implicated in inflammation or the immune response. The inflammatory component of MPS IVA results from GAG accumulation within tissues that activate Toll-like receptors and inflammation pathways [38]. Anti-inflammatory drugs can inhibit the activity of altered cytokines and interleukins in MPS IVA [4]. In our study population, expression levels of certain dysregulated ILs (e.g., IL1 and IL1β) were not corrected after ERT. ILs have been described by other authors as possible candidate biomarkers for MPS IVA [23]. The combination of anti-inflammatory drugs (infliximab) and ERT has been tested in rats with MPS IVA, in which analysis of cultured articular chondrocytes showed that this therapy restored collagen IIA1 expression and reduced the expression of apoptotic markers [39]. However, this treatment only suppressed hyperplasia of synovial cells. Micro-computed tomography (CT) analysis revealed no significant positive effects on bone microarchitecture and no histological improvement in bone growth plates was observed. In our study, ERT resulted in only partial correction of the expression of dysregulated proteins implicated in the immune response. It is not clear if the possible alterations in the immune response that can affect patients with MPS IVA are due to the neutralizing antibodies that they produce against the ERT enzyme and there are no studies to clarify this.

Our quantitative analysis examined potential alterations in the expression of 215 proteins included in the SWATH library. Using this method, we detected proteins that showed changes in expression all other groups relative to healthy controls. In line with the findings of the qualitative analysis, several of the dysregulated proteins identified using this approach are involved in bone metabolism. Two such proteins, VTNC and FINC, participate in remodeling of the extracellular matrix (ECM) and in osseous integration mechanisms [40]. VTNC is a glycoprotein that circulates in blood and is also found in the ECM. This protein can bind GAGs, collagen, plasminogen, and the urokinase receptor. VTNC regulates the proteolytic degradation of this matrix. Moreover, VTNC binds to the complement system, heparin, and thrombin-antithrombin III complexes, facilitating the regulation of the immune response and clot formation [41].

Fetuin A (*FETUA*) [42] is a family member structurally related to the cystatin-like protein domain. Cystatin can inhibit papain, caspases, and cathepsins. *FETUA* inhibits insulin receptor autophosphorylation and plays a key role in insulin-dependent metabolism. Fetuin-A is also involved in osteogenesis and bone resorption and in the control of calcium salt precipitation in blood [43,44]. This protein accumulates in the bone matrix and inhibits transforming growth factor, necessary for bone mineralization [39]. Fetuin-A is also an anti-inflammatory mediator that participates in macrophage deactivation. Decreases in plasma levels of this protein may occur in response to inflammatory processes, such as those that occur in MPS IVA [45]. A proteomic analysis of pig bone samples from Procopio et al. presents many of the proteins located in different regions of the bone, such as in the proximal area of the tibia where the FA10, VTNC, SAMP, CRP, and FETUA proteins were found [46]. The remaining protein expression found in the study was located in the metatarsus, the midshaft of the tibia and the proximal femur. The authors concluded that three proteins, A1T1, chromogranin A, and FETUA, could be considered biological age markers in the tibia [46]. To identify candidate protein biomarkers of MPS IVA, Martell et al. performed quantitative multiplexed assays using plasma samples from 78 MPS IVA patients and 58 healthy controls [33]. The authors identified three candidate proteins: alpha-1-antitrypsin, lipoprotein(a), and serum amyloid. In our analysis, we found that A1T1 expression was increased in untreated MPS IVA patients and decreased in ERT-treated patients.

Another protein identified in our study was serum amyloid A 4 (SAA4). The SAA protein family has been described as important clinical markers of inflammation and acts as a precursor of elements that are involved in secondary reactive amyloidosis [47]. Moreover, SAA family proteins are known to contribute to cellular cholesterol homeostasis, modulate intracellular calcium levels, and promote signaling cascades [48,49,50]. SAA proteins also participate in lipid transport and metabolism and are implicated in atherosclerosis. SAA proteins share several common functions with cytokines, including cell–cell communication and feedback in inflammatory, immunological, neoplastic, and protective pathways [51]. Retinol-binding protein 4 (RET4) is the main transporter of retinol in the bloodstream, whose metabolites may contribute to osteogenesis, chondrogenesis, and adipogenesis. RET4 is mainly produced in the liver (but is also expressed in several extrahepatic tissues, including the limbs), where it mobilizes hepatic retinol stores [52].

Another protein of interest detected in the present study is CLUS. This protein helps to prevent cell stress-induced apoptosis and participates in the aggregation of blood plasma proteins [53,54,55,56,57], along with other proteins such as APP and APOC2. CLUS can also inhibit the formation of amyloid fibrils.

### Study Limitations

The SWATH method is a specific, reproducible, and sensitive approach that allows relative or absolute quantification of proteins. However, some limitations of this approach should be noted. First, the number of proteins identified/quantified is primarily limited by the composition of the spectral library. In this study, the library was generated from a pool of proteins for each group of study, and consisted of 215 proteins, each with a FDR < 1%. In contrast to the selected reaction monitoring (SRM) technique, in which only three transitions are quantified, the SWATH technique quantifies seven transitions are for each peptide. Therefore, 10 peptides per protein and seven fragments per peptide must be identified in order to extract the peak areas necessary for quantification. Despite increasing the precision of the technique, this aspect also constitutes a limitation, as proteins in the library for which less than 10 peptides are identified will not be quantified.

Another limitation of our study is that normalization of the expression of proteins in plasma does not necessarily correlate with normalization in chondrocytes in the vascular region. Enzyme is easily taken up by hepatic proteins, restoring dysregulated metabolic pathways in these cells while the infused enzyme circulates in the blood. It is critical to understand whether the blood/plasma proteins are dysregulated in bone and cartilage in MPS IVA patients in order to identify potential diagnostic biomarkers of disease severity. Moreover, when a bone-penetrating drug becomes available, surrogate disease, progression disease, or treatment outcome biomarker will be essential.

The selected candidate proteins must be studied using techniques that are used daily in clinical practice to demonstrate their validity. However, the low number of patients with this disease makes this analysis difficult.

## 4. Materials and Methods

### 4.1. Study Workflow

The graphical abstract depicts the workflow used for the qualitative (DDA-LC-MS/MS) and quantitative (SWATH-MS) analysis of proteins in plasma samples from healthy controls and MPSIVA patients (treated or untreated) (Figure 14).

### 4.2. Samples

After obtaining written informed consent, blood samples from MPS IVA patients were acquired for proteomic analysis of plasma from three Spanish centers of reference for MPS IVA: eight from untreated MPS IVA patients; and five from MPS IVA patients who had been undergoing ERT for a mean duration of 7 years, and from whom samples were taken 24 h before (ERT-a group) and 24 h after (ERT-b group) one of their weekly ERT infusions. All patients presented the classical phenotype as is shown in the previous study [32]. Another 6 blood samples were acquired from 6 healthy donors (Figure 15). This study was approved (22 May 2017) by the local Ethics Committee of Santiago-Lugo (Reference number 2017/298).

### 4.3. Protein Extraction

Plasma samples were depleted in order to eliminate the most abundant proteins prior to the proteomic analysis. Leukocytes were sonicated to rupture the cell membrane and centrifuged for 10 min at 10,000 rpm and 4 °C. Protein extracts from the supernatant were recovered and subsequently frozen at −20 °C.

### 4.4. Enzyme Activity Test

An enzyme activity test was used to analyze the GALNS activity in plasma samples [58]. Enzymatic activity in samples from ERT-a patients was lower than reference values (1.8–20 nmol/h/mg protein) established by the laboratory to diagnose MPSIVA.

#### 4.5. Proteomic Analysis

Proteomic analysis was performed as described previously by our group [32]. Plasma samples were first depleted as previously described [59]. A restrictive analysis was performed to select only proteins with a *p*-value < 0.05 and a fold change >1.5 as possible candidate biomarkers. A heat map was generated using http://www.heatmapper.ca/expression/, accessed on 1 May 2021.

## 5. Conclusions

Morquio A is a rare disease. Diagnosis can be complicated in the absence of clinical signs and symptoms, and bone impact therapy remains undeveloped. The discovery of effective biomarkers for rare diseases is essential to improve diagnosis and treatment. Based on our findings, we propose plasma levels of FETUA, VINT, A1AT, and CLUS as potential candidate biomarkers for MPS IVA. These proteins may influence bone and cartilage metabolism but more studies are needed. Other proteins identified in this study appear to provide an indication of the effects produced as a result of bone alteration. In particular, further investigation of the role of A1AT in bone is warranted, as several studies have reported dysregulation of this protein in MPS IVA.

## Figures and Tables

**Figure 1 ijms-22-06165-f001:**
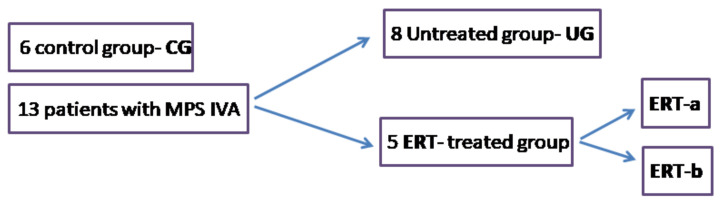
Study groups: controls and MPS IVA patients.CG, control group; MPS IVA, mucopolysaccharidosis type IVA; ERT-a, MPS IVA patients sampled before ERT; ERT-b, MPS IVA patients sampled 24 h after ERT.

**Figure 2 ijms-22-06165-f002:**
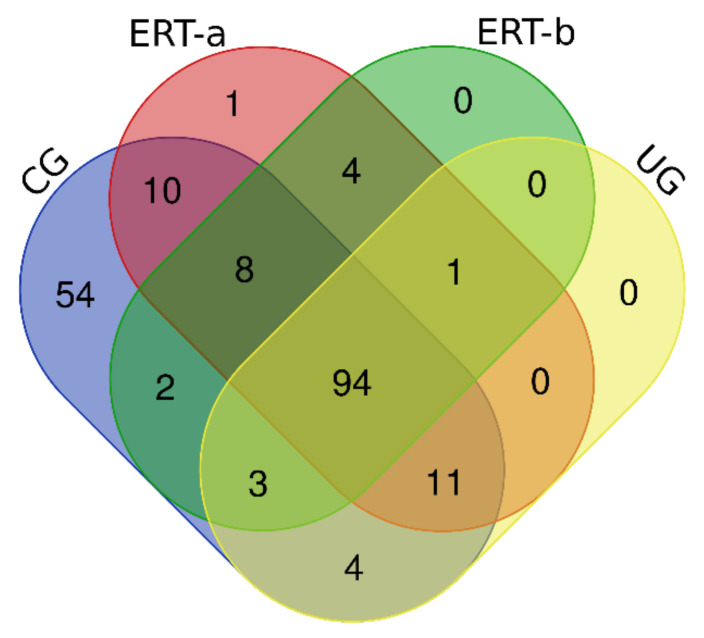
Venn diagram showing the distribution of proteins identified in the qualitative proteomic analysis across each of the 4 study groups.

**Figure 3 ijms-22-06165-f003:**
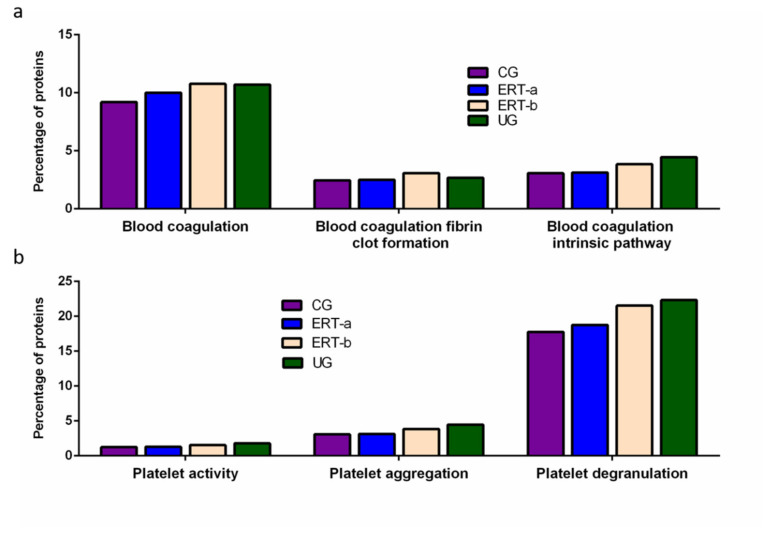
Functional enrichment analysis using FunRich. Proteins identified in each the different groupswere submitted to molecular functional analysis according to their involvement inhemostasis (**a**) and platelet function (**b**). CG, control group; ERT-a, patients sampled before ERT; ERT-b, patients sampled 24 h after ERT; UG, untreated group. Proteins listed for qualitative proteomic analysis appear in Appendix A.

**Figure 4 ijms-22-06165-f004:**
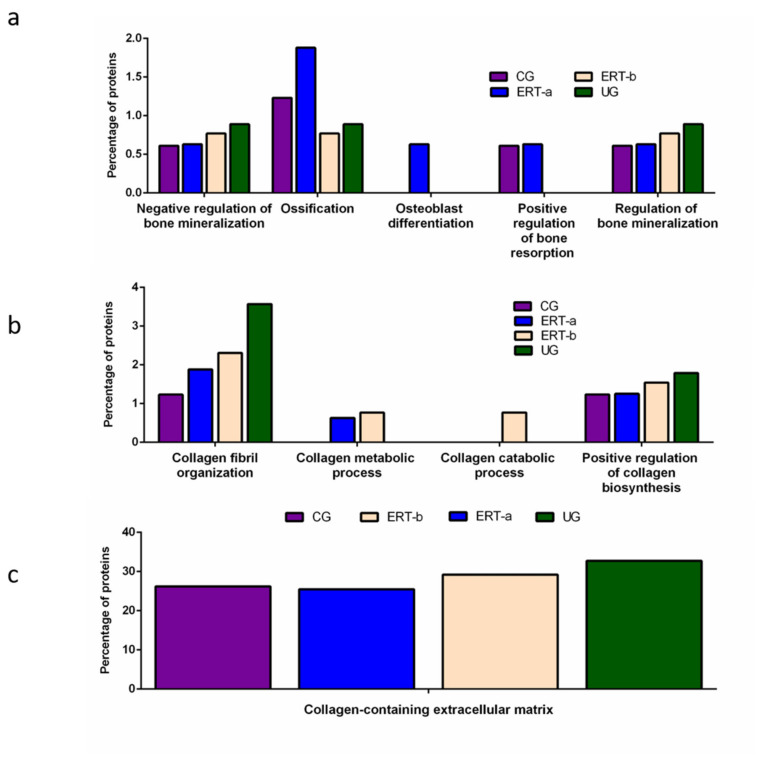
Functional enrichment analysis using FunRich. Proteins identified in each the different groups were submitted to molecular functional analysis according to their involvement in: (**a**) processes that occur at the bone level; (**b**) collagen function (activity/processing/biosynthesis); (**c**) collagen-containing extracellular matrix. CG, control group; ERT-a, patients sampled before ERT; ERT-b, patients sampled 24 h after ERT; UG, untreated group. Proteins included in this analysis, identified in the qualitative proteomic analysis, are listed in the Appendix A.

**Figure 5 ijms-22-06165-f005:**
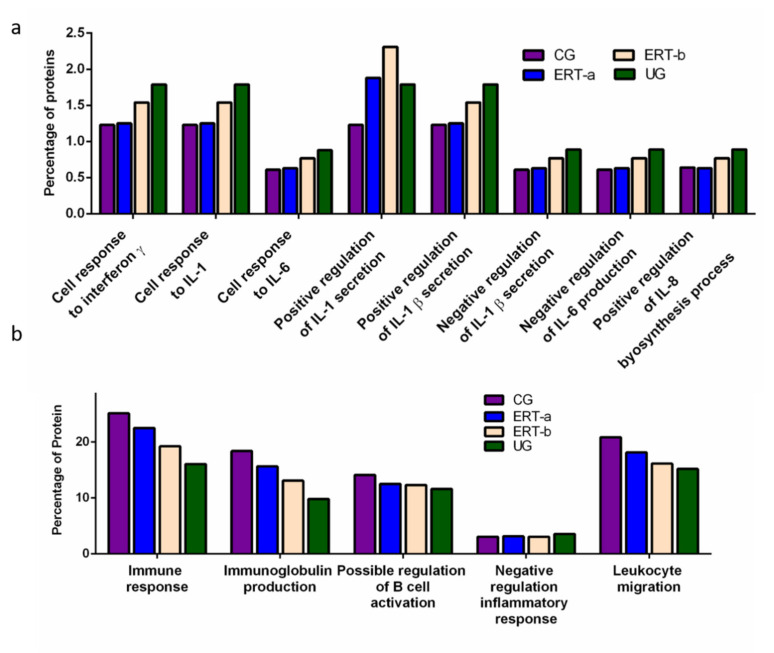
Functional enrichment analysis using FunRich. Proteins identified in each the different groups were submitted to molecular functional analysis according to their involvement in the immune response: proteinsrelated tointerleukins (ILs) IL-1, IL-1β, IL-6, IL-8, and interferon γ (**a**); proteins involved in immunoglobulin production, positive regulation of B-cell activation, negative regulation of the inflammatory response, and leukocyte migration (**b**).

**Figure 6 ijms-22-06165-f006:**
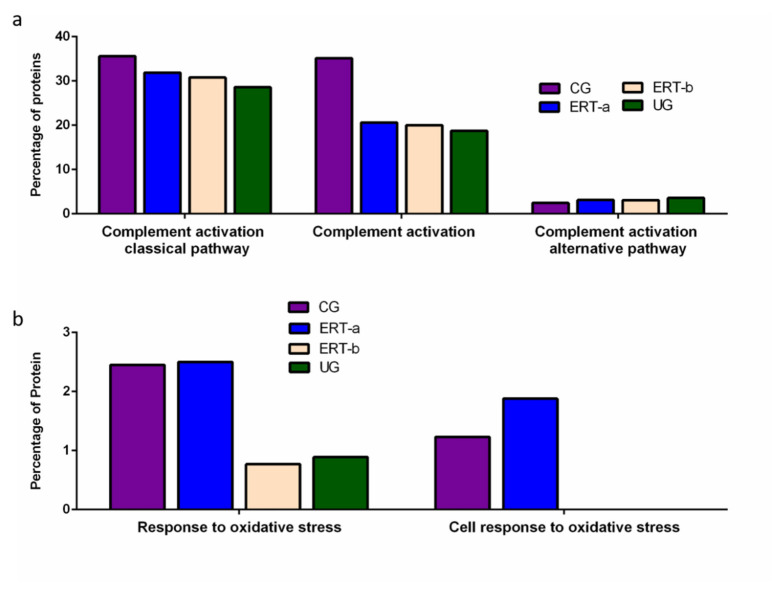
Functional enrichment analysis using FunRich. Proteins identified in each of the different groups were submitted to molecular functional analysis according to their involvement in immune processes: proteins involved in the classical or alternative complement activation pathways (**a**); proteins involved in the response to oxidative stress (**b**).

**Figure 7 ijms-22-06165-f007:**
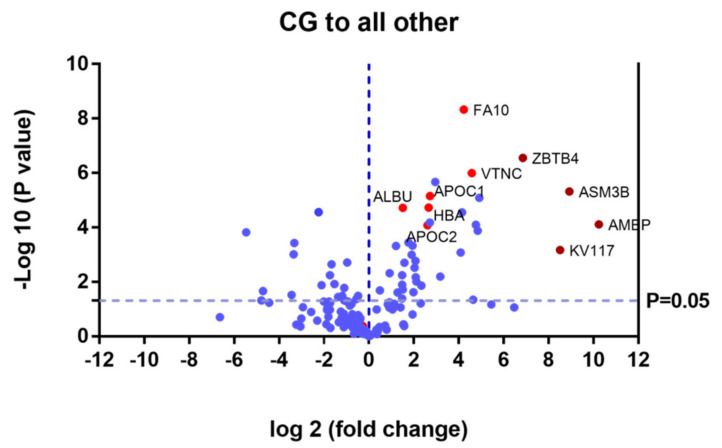
Volcano plot showing the results of the SWATH analysis of plasma protein expression. Data are represented as the results obtained for the control group relative to all other groups (UG, ERT-a, and ERT-b). Graph shows the negative base 10 log of the *p*-value plotted against the base 2 log of the fold change for each protein included in the SWATH analysis. Changes were considered significant at *p* < 0.05 and fold change >1.5 compared with control group.

**Figure 8 ijms-22-06165-f008:**
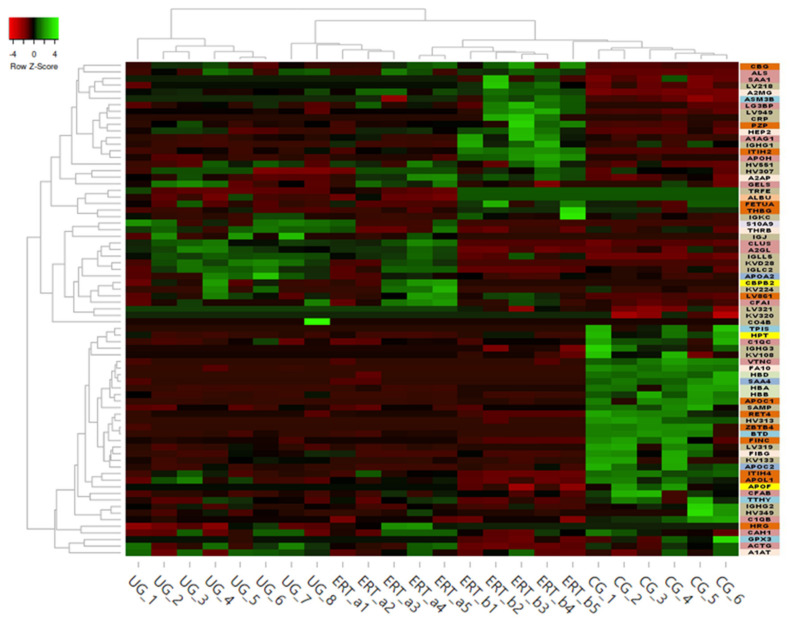
Heatmap depicting the expression profiles of different proteins (also shown in Table 2, Table 3 and Table 4) as determined by SWATH quantitative analysis. The relative abundance of each protein is reflected by the colors in the heatmap, based on the z-score of the protein’s normalized peak area. The dendrogram represents the biological replicates of the UG, ERT-a, ERT-b, and CG groups. Proteins, listed in the sidebar on the right, are color coded according to their function, as follows: apolipoprotein-related proteins, dark green; blood coagulation, purple; calcium binding, dark orange; carrier proteins, light green; cytoskeletal proteins, light pink; immune system, grey; metabolic processes, yellow; metabolic interconversion, light blue; other functions, white; protein binding, pink; transport, light grey.

**Figure 9 ijms-22-06165-f009:**
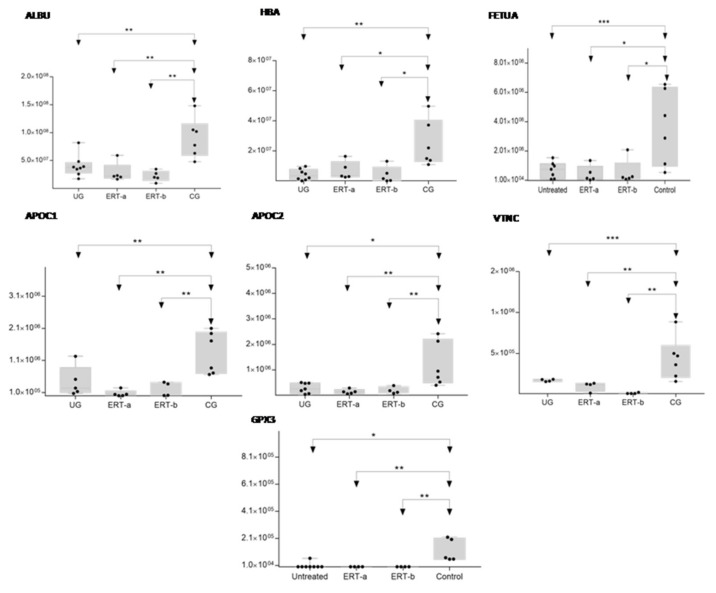
Box plot depicting expression levels of the6 most relevant plasma proteins (ALBU, HBA, FETUA, APOC1, APOC2, VTNC, and GPX3) for which downregulation was observed in the UG and ERT-a and ERT-b groups relative to the control group (CG). Each data point represents the median protein expression value obtained in an individual sample. The line inside the box represents the median of all values obtained. The upper and lower limits of the box represent the first and third quartiles. Whiskers represent the minimum and maximum values within 1.5 times the interquartile range. Any data points not included between the whiskers are considered outliers. * *p* < 0.05; ** *p* < 0.01; *** *p* < 0.001.

**Figure 10 ijms-22-06165-f010:**
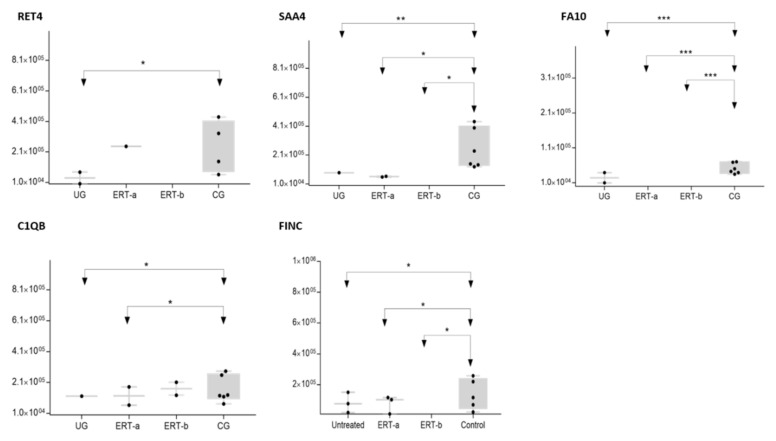
Box plot depicting expression levels of the 5 most relevant plasma proteins (RET4, SAA4, FA10, FINC, and C1QB) for which very low expression was observed in the UG and ERT-a and ERT-b groups relative to the control group (CG). Each data point represents the median value obtained in an individual sample. The line inside the box represents the median of all values obtained. The upper and lower limits of the box represent the first and third quartiles. Whiskers represent the minimum and maximum values within 1.5 times the interquartile range. Any data points not included between the whiskers are considered outliers. * *p* < 0.05; ** *p* < 0.01; *** *p* < 0.001.

**Figure 11 ijms-22-06165-f011:**
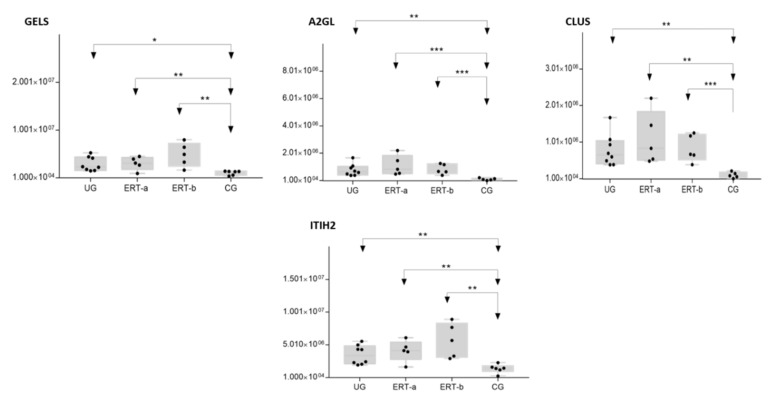
Box plot depictingthe4 plasma proteins (A2GL, CLUS, GELS, ITIH2) with the highest expression levels in the UG and ERT-a and ERT-b groups relative to the control group (CG). Each data point represents the median value obtained in an individual sample. The line inside the box represents the median of all values obtained. The upper and lower limits of the box represent the first and third quartiles. Whiskers represent the minimum and maximum values within 1.5 times the interquartile range. Any data points not included between the whiskers are considered outliers. * *p* < 0.05; ** *p* < 0.01; *** *p* < 0.001.

**Figure 12 ijms-22-06165-f012:**
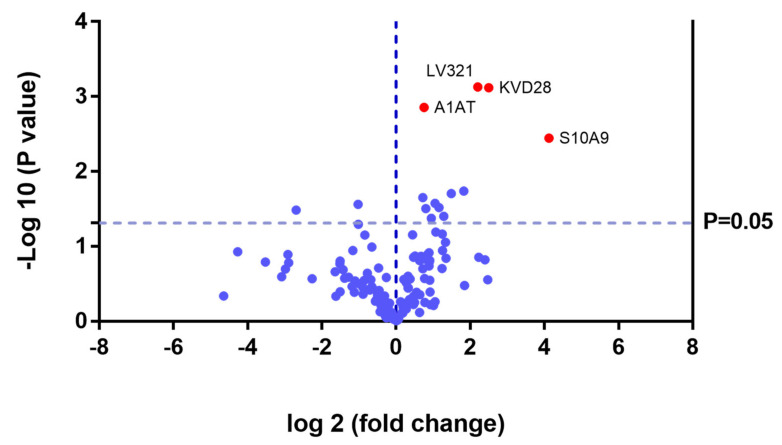
Volcano plot showing the results of the SWATH analysis of plasma protein expression. Data are represented as the results obtained for the untreated group (UG) relative to all other groups (CG, ERT-a, and ERT-b). Graph shows the negative base 10 log of the *p*-value plotted against the base 2 log of the fold change for each protein included in the SWATH analysis. Changes were considered significant at *p* < 0.05 and fold change >1.5.

**Figure 13 ijms-22-06165-f013:**
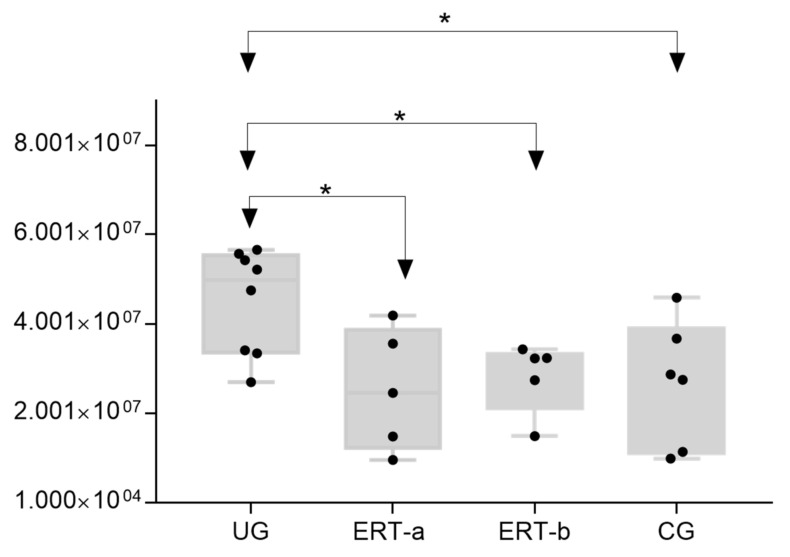
Box plot depicting ATA1 levels in each of the study groups. Each data point represents the median value obtained in an individual sample. The line inside the box represents the median of all values obtained. The upper and lower limits of the box represent the first and third quartiles. Whiskers represent the minimum and maximum values within 1.5 times the interquartile range. Any data points not included between the whiskers are considered outliers. * *p* < 0.05.

**Figure 14 ijms-22-06165-f014:**
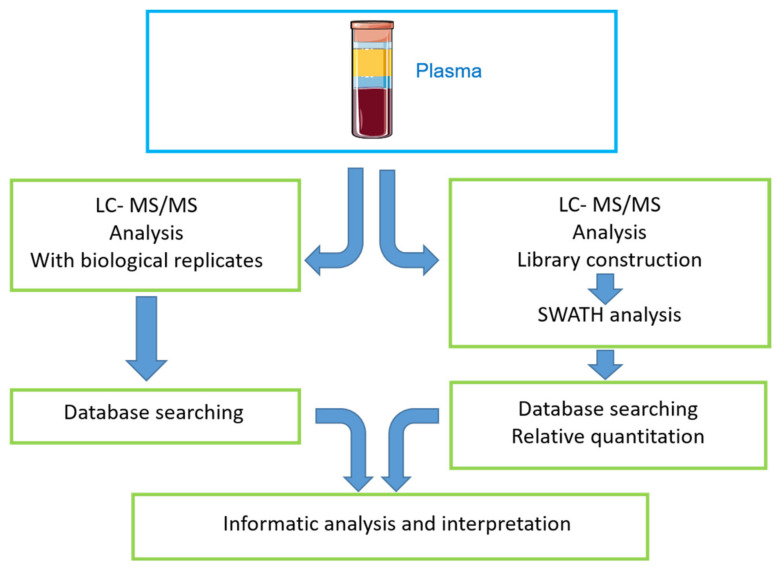
Graphical abstract showing study workflow. Plasma samples were acquired from 3 distinct groups: untreated MPS IVA patients; MPS IVA patients that received ERT (from whom samples were taken 1 day before and 1 day after ERT), and healthy blood donors. Blood was separated to obtain plasma, the samples were lysed, and proteomic analyses were performed. On completion of these two distinct proteomic analyses, a bioinformatics analysis was performed to obtain more information about the identified proteins.

**Figure 15 ijms-22-06165-f015:**
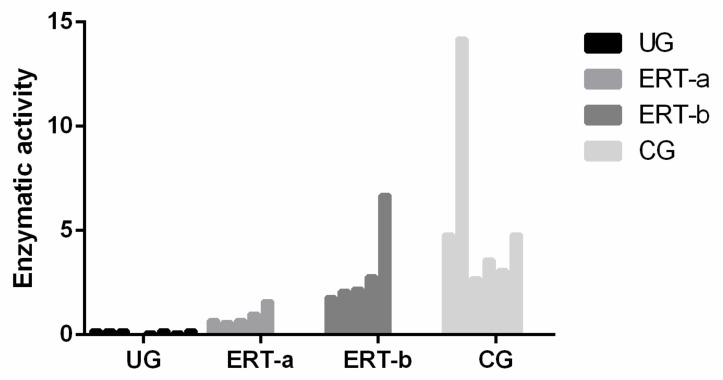
Enzyme activity of GALNS. CG, control group; ERT-a, MPS IVA patients sampled 1 day before ERT; ERT-b, MPS IVA patients sampled 24 h after ERT; UG, untreated group.

**Table 1 ijms-22-06165-t001:** Changes in plasma protein expression detected by SWATH in any of the patient groups (UG, ERT-a, or ERT-b) relative to the control group.

Group Comparison	Proteins with Altered Expression*	Upregulated Proteins	Downregulated Proteins
UG vs CG	14	12	2
ERT-a vs CG	8	5(2 only in ERT-a)	1
ERT-b vs CG	14	10(2 only in ERT-b)	1

CG, control group; ERT-a, patients sampled before ERT; ERT-b, patients sampled 24 h after ERT; UG, untreated group. * *p* < 0.05 and fold change >1.5 compared with the control group.

**Table 2 ijms-22-06165-t002:** Changes in plasma protein expression detected in all patient groups (UG, ERT-a, and ERT-b) relative to the control group. Protein expression is considered altered in cases in which *p* < 0.05 and the fold change >1.5 (upregulated proteins) or <0.6 (downregulated proteins).

Uniprot Code	Protein Code	Protein Name	*p*-Value	Fold Change
**Downregulated in UG, ERT-a, and ERT-b**				
P01860	IGHG3	Immunoglobulin heavy constant gamma 3	3.30 × 10^−5^	Only CG
P02741	CRP	C-reactive protein	8.93 × 10^−3^	Only CG
P02760	AMBP	Protein AMBP	7.75 × 10^−5^	0.0008
Q92485	ASM3B	Acid sphingomyelinase-like phosphodiesterase 3b	4.80 × 10^−6^	0.0021
P01599	KV117	Immunoglobulin kappa variable 1–17	6.69 × 10^−4^	0.0027
Q9P1Z0	ZBTB4	Zinc finger and BTB domain-containing protein 4	2.81 × 10^−7^	0.0086
P01780	HV307	Immunoglobulin heavy variable 3–7	8.23 × 10^−6^	0.0333
P22352	GPX3	Glutathione peroxidase 3	1.31 × 10^−4^	0.0349
P35542	SAA4	Serum amyloid A-4 protein	8.01 × 10^−5^	0.0368
P01766	HV313	Immunoglobulin heavy variable 3–13	4.50 × 10^−2^	0.0402
P04004	VTNC	Vitronectin	1.00 × 10^−6^	0.0417
P00742	FA10	Coagulation factor X	4.71 × 10^−9^	0.0535
P02042	HBD	Hemoglobin subunit delta	2.79 × 10^−5^	0.0565
A0A075B6J9	LV218	Immunoglobulin lambda variable 2–18	8.30 × 10^−4^	0.0588
P02753	RET4	Retinol-binding protein 4	6.31 × 10^−3^	0.1103
P06312	KV401	Immunoglobulin kappa variable 4–1	2.12 × 10^−6^	0.1292
P02654	APOC1	Apolipoprotein C-I	7.01 × 10^−6^	0.1515
P02765	FETUA	Alpha-2-HS-glycoprotein	6.60 × 10^−5^	0.1527
P69905	HBA	Hemoglobin subunit alpha	1.86 × 10^−5^	0.1585
P02655	APOC2	Apolipoprotein C-II	8.27 × 10^−5^	0.1642
P00915	CAH1	Carbonicanhydrase 1	1.37 × 10^−2^	0.1984
P02751	FINC	Fibronectin	9.79 × 10^−3^	0.2304
A0A0B4J1Y8	LV949	Immunoglobulin lambda variable 9–49	6.70 × 10^−3^	0.2353
P02746	C1QB	Complement C1q subcomponent subunit B	1.68 × 10^−3^	0.2381
P00738	HPT	Haptoglobin	3.00 × 10^−3^	0.2445
P01857	IGHG1	Immunoglobulin heavy constant gamma 1	2.37 × 10^−2^	0.2519
P01834	IGKC	Immunoglobulin kappa constant	4.67 × 10^−4^	0.2591
P68871	HBB	Hemoglobin subunit beta	1.01 × 10^−3^	0.2681
P01859	IGHG2	Immunoglobulin heavy constant gamma 2	3.60 × 10^−4^	0.2959
P29622	KAIN	Kallistatin	1.99 × 10^−3^	0.3344
P02743	SAMP	Serum amyloid P-component	1.91 × 10^−2^	0.3509
P02768	ALBU	Serum albumin	1.92 × 10^−5^	0.3509
P02787	TRFE	Serotransferrin	4.11 × 10^−2^	0.3571
P02749	APOH	Beta-2-glycoprotein 1	1.26 × 10^−2^	0.3584
P02679	FIBG	Fibrinogen gamma chain	5.71 × 10^−3^	0.3597
P02763	A1AG1	Alpha-1-acid glycoprotein 1	2.43 × 10^−2^	0.4115
P05543	THBG	Thyroxine-binding globulin	4.82 × 10^−4^	0.4348
**Upregulated in UG, ERT-a, and ERT-b**				
P01619	KV320	Immunoglobulin kappa variable 3–20	0.0005	Not found CG
P80748	LV321	Immunoglobulin lambda variable 3-21	0.0179	Not found CG
B9A064	IGLL5	Immunoglobulin lambda-like polypeptide 5	0.0002	44.0529
P01615	KVD28	Immunoglobulin kappa variable 2D–28	0.0474	27.4725
P0DOY2	IGLC2	Immunoglobulin lambda constant 2	0.0217	26.1780
P35858	ALS	Insulin-like growth factor-binding protein complex acid labile subunit	0.0299	10.8460
P10909	CLUS	Clusterin	0.0010	10.1523
P02750	A2GL	Leucine-rich alpha-2-glycoprotein	0.0004	9.9800
P01023	A2MG	Alpha-2-macroglobulin	0.0000	4.7170
P00734	THRB	Prothrombin	0.0132	4.3178
P06396	GELS	Gelsolin	0.0057	3.3223
P19823	ITIH2	Inter-alpha-trypsin inhibitor heavy chain H2	0.0023	3.1536
Q96IY4	CBPB2	Carboxypeptidase B2	0.0121	2.8977
Q08380	LG3BP	Galectin-3-binding protein	0.0361	2.5880
P08185	CBG	Corticosteroid-binding globulin	0.0325	2.2763
A0A0A0MS15	HV349	Immunoglobulin heavy variable 3–49	0.0482	2.1418
A0A0C4DH38	HV551	Immunoglobulin heavy variable 5–51	0.0165	2.1345

Only CG: These proteins were not detected in theUG, ERT-a, or ERT-b groups, and were only found in CG. Not found CG: these proteins were not found in CG, and were only detected in the UG, ERT-a, or ERT-b groups.CG, control group; ERT-a, patients sampled before ERT; ERT-b, patients sampled 24 h after ERT; UG, untreated group.

**Table 3 ijms-22-06165-t003:** Changes in protein expression in the untreated group relative to all other groups. Protein expression is considered altered in cases in which *p* < 0.05 and the fold change >1.5 (upregulated proteins) or <0.6 (downregulated proteins).

Uniprot Code	Protein Code	Protein Name	*p*-Value	Fold Change
**Upregulated in Untreated Group**				
P06702	S10A9	Protein S100-A9	0.0036	17.4468
P01615	KVD28	Immunoglobulin kappa variable 2D-28	0.0008	5.6619
P80748	LV321	Immunoglobulin lambda variable 3-21	0.0007	4.6098
P01594	KV133	Immunoglobulin kappa variable 1-33	0.0182	3.5490
P02652	APOA2	Apolipoprotein A-II	0.0197	2.8151
P01591	IGJ	Immunoglobulin J chain	0.0395	2.4479
P63261	ACTG	Actin, cytoplasmic 2	0.0304	2.2374
P00734	THRB	Prothrombin	0.0265	2.0770
B9A064	IGLL5	Immunoglobulin lambda-like polypeptide 5	0.0420	1.9341
P01714	LV319	Immunoglobulin lambda variable 3-19	0.0313	1.7496
P01009	A1AT	Alpha-1-antitrypsin	0.0014	1.6875
P02747	C1QC	Complement C1q subcomponent subunit C	0.0223	1.6522
**Downregulated in Untreated Group**				
P02746	C1QB	Complement C1q subcomponent subunit B	0.0329	0.1549
P08697	A2AP	Alpha-2-antiplasmin	0.0275	0.4938

**Table 4 ijms-22-06165-t004:** Changes in protein expression in the ERT-a or ERT-b groups relative to all other groups. Protein expression is considered altered in cases in which *p* < 0.05 and the fold change >1.5 (upregulated proteins) or <0.6 (downregulated proteins).

**Uniprot Code**	**Protein Code**	**Protein Name**	***p*** **-Value**	**Fold Change**
**Upregulated in ERT-a**				
P0DJI8	SAA1	Serum amyloid A-1 protein	0.0487	Only ERT-a
Q13790	APOF	Apolipoprotein F	0.0487	Only ERT-a
A0A0C4DH68	KV224	Immunoglobulin kappa variable 2-24	0.0109	6.1248
P05156	CFAI	Complement factor I	0.0065	5.2409
A0A075B6I0	LV861	Immunoglobulin lambda variable 8-61	0.0386	3.1547
P43251	BTD	Biotinidase	0.0301	2.8996
P04196	HRG	Histidine-rich glycoprotein	0.0224	1.5688
**Downregulated in ERT-a**				
P02747	C1QC	Complement C1q subcomponent subunit C	0.0465	0.5069
**Uniprot Code**	**Protein Code**	**Protein Name**	***p*** **-Value**	**Fold Change**
**Upregulated in ERT-b**				
P60174	TPIS	Triosephosphateisomerase	0.0487	Only ERT-b
O00391	QSOX1	Sulfhydryl oxidase 1	0.0487	Only ERT-b
A0A0C4DH67	KV108	Immunoglobulin kappa variable 1-8	0.0028	Only ERT-b
P0C0L5	CO4B	Complement C4-B	0.0029	17.9592
O14791	APOL1	Apolipoprotein L1	0.0480	2.2700
P20742	PZP	Pregnancyzone protein	0.0418	2.1171
P06396	GELS	Gelsolin	0.0080	2.0649
P00751	CFAB	Complement factor B	0.0184	2.0058
P19823	ITIH2	Inter-alpha-trypsin inhibitor heavy chain H2	0.0093	1.9167
P01023	A2MG	Alpha-2-macroglobulin	0.0093	1.8574
P08697	A2AP	Alpha-2-antiplasmin	0.0298	1.8165
Q14624	ITIH4	Inter-alpha-trypsin inhibitor heavy chain H4	0.0389	1.7428
P05546	HEP2	Heparin cofactor 2	0.0421	1.6111
**Downregulated in ERT-b**				
P02766	TTHY	Transthyretin	0.0224	0.2358

Only ERT-a: these proteins wereupregulated in the ERT-a group with an FC value of infinity, meaning that they wereabsentin all other groups (UG, CG, and ERT-b). Only ERT-b: These proteins wereupregulated in the ERT-b group with aFC value of infinity, meaning that they wereabsent in all other groups (UG, CG, and ERT-a).

## Data Availability

The data presented in this study are available in this article and this Supplementary Material.

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
