# Peer review of "Plasma Proteomic Analysis in Morquio A Disease"

_ijms, 2021, doi:10.3390/ijms22116165_

Round 1

Reviewer 1 Report

Álvarez et al. aimed to identify biomarkers in serum of MPS IVA patients. They wanted to identify biomarkers for the disease itself and for the treatment success of ERT. They were especially aiming for biomarkers to evaluate the therapy success of bone lesions. They performed qualitative and quantitative proteomic analysis and summarize their data in this manuscript.

It is important to identify biomarkers, especially for monitoring the treatment success, however, the manuscript needs significant improvement. The data are not presented in an understandable way. Furthermore, the potential biomarkers that were identified during the proteomic screen should be confirmed with a different method (e.g. ELISA, Western blot, enzyme activity assay). In general, in order to be used as a biomarker, the proteins of interest should be measurable with an easy routine method.

Specific comments:

  • What is the difference between untreated MPS IVA patients and those before ERT? Isn’t “before ERT” the same as untreated? Or have those patients undergone ERT before? If yes, this should be made clear (i.e. since when and how often did they receive ERT)
  • On page 3 you talk about samples after 7 days after ERT. What does this mean? According to the methods and Figure 1, there are only samples 24 h past ERT. If plasma was obtained also after 7 days, the data should be included in the manuscript.
  • The results part about the qualitative analysis is presented in a very poor way. Figures 2,3,5,6 should be presented in a different way, e.g. in a single table. What about error bars and statistics? Table 1 can be omitted or transferred to the supplementary section. What is the purpose of Figure 4?
  • Section 2.2 containing the quantitative analysis for sure is more interesting, but for me it was impossible to understand why certain proteins were chosen and others not. More specifically: Table 2 states that 2 proteins are downregulated in UG and 1 protein in ERT-a and –b, but Table 3 contains a long list of downregulated proteins. What are table 2 and 3 referring to? Why are the numbers so different? What does “2 only in ERT-a” mean in Table 2? In the text, GPX3 is emphasized, but somehow did not make it into the group of biomarkers. Why not? GPX3 activity can be measured easily and might therefore be useful as a biomarker.
  • Figure 8 and 9: These figures show the “most relevant plasma proteins”, but why are they most relevant? According to Table 3, they neither have the highest fold change nor the highest significance value?
  • Figure 11 and text on the page before state that in the untreated group 4 proteins are significantly different from all other groups. Wouldn’t those be the best biomarkers for treatment success? However, those 4 proteins are different from the 4 proteins that were in the end chosen as putative biomarkers: in Figure 11 these are LV321, KVD28, A1AT and S10A9, while in the end FETUA, VINT, A1AT and CLUS were postulated as biomarkers?
  • How does Table 4 relate to Figure 11, because in Table 4 a lot more than 4 proteins are listed?
  • I understand that A1AT might be a good candidate. However, this should be confirmed with another method.
  • Table 5: What is the significance of this table? If this lists proteins that differ between ERT-a/b and all other groups, does this mean healthy probands and untreated patients show the same expression?

Author Response

REVIEWER 1.

Álvarez et al. aimed to identify biomarkers in serum of MPS IVA patients. They wanted to identify biomarkers for the disease itself and for the treatment success of ERT. They were especially aiming for biomarkers to evaluate the therapy success of bone lesions. They performed qualitative and quantitative proteomic analysis and summarize their data in this manuscript.

It is important to identify biomarkers, especially for monitoring the treatment success, however, the manuscript needs significant improvement. The data are not presented in an understandable way. Furthermore, the potential biomarkers that were identified during the proteomic screen should be confirmed with a different method (e.g. ELISA, Western blot, enzyme activity assay). In general, in order to be used as a biomarker, the proteins of interest should be measurable with an easy routine method.

Specific comments:

  • What is the difference between untreated MPS IVA patients and those before ERT? Isn’t “before ERT” the same as untreated? Or have those patients undergone ERT before? If yes, this should be made clear (i.e. since when and how often did they receive ERT)

ANSWER: The untreated MPS IVA groups consists of patients who never received treatment. The ERT groups consist of samples that were acquired from MPS IVA patients undergoing ERT for a mean duration of 7 years, from whom samples were acquired 24 hours before (ERT-a) and again 24 hours after (ERT-b)one of their weekly ERT infusions.

For greater clarify regarding the provenance of the samples in each group, the Results section has been edited as follows:

“Plasma samples were acquired from 13 MPS IVA patients and 6 controls and classified as follows: healthy controls; untreated MPS IVA patients; and MPS IVA patients who had been receiving weekly ERT infusions for a mean of 7 years. From these ERT-treated patients, samples were acquired 24 h before (ERT-a) and 24 h after (ERT-b) one of their weekly infusions (Fig. 1). The demographic features of the participating MPS IVA patients have been previously described [33].”

  • On page 3 you talk about samples after 7 days after ERT. What does this mean? According to the methods and Figure 1, there are only samples 24 h past ERT. If plasma was obtained also after 7 days, the data should be included in the manuscript.

ANSWER: This is indeed confusing. As explained in the previous answer, the ERT-a samples were acquired from patients who had been receiving weekly ERT infusions for a mean of 7 years. Thus, ERT-a samples correspond to those collected 6 days after the preceding infusion, and 24 hours before the next infusion. This is now made clear in the Results section, as indicated in the quoted text above.

  • The results part about the qualitative analysis is presented in a very poor way. Figures 2,3,5,6 should be presented in a different way, e.g. in a single table. What about error bars and statistics? Table 1 can be omitted or transferred to the supplementary section. What is the purpose of Figure 4?

ANSWER: We agree with the reviewer's comment regarding Figures 2,3,5, and 6. These figures were generated using FunRich (Functional Enrichment analysis tool. http://www.funrich.org/),aprogram designed in 2015to perform functional analyses of qualitative proteomic data.

Pathan M, Keerthikumar S, Ang CS, Gangoda L, Quek CY, Williamson NA, Mouradov D, Sieber OM, Simpson RJ, Salim A, Bacic A, Hill AF, Stroud DA, Ryan MT, Agbinya JI, Mariadason JM, Burgess AW, Mathivanan S. FunRich: An open access standalone functional enrichment and interaction network analysis tool. Proteomics. 2015;15(15):2597-601.

Benito-Martin A, Peinado H. FunRich proteomics software analysis, let the fun begin! Proteomics. 2015;15(15):2555-6.

This program is used extensively by the scientific community to perform functional analyses of qualitative proteomic data between samples, given the differences in the percentage protein expression found in UNIPROT databases. Moreover, the program can provide information about cellular behavior, molecular function, biological processes, and protein domains.

This type of analysis is not a statistical analysis, but rather detects differences in protein expression in the context of the function of interest. that the impossibility of performing a statistical analysis is a limitation of the program used for this functional analysis.

Moreover, the proteins depicted in these figures were selected based on alterations previously described in MPS IVA. In almost all cases, our findings were in line with that which is already known about the pathology (e.g. the increase in the levels of specific interleukins in MPS IVA patients).

In accordance with the reviewer’s suggestions, Figure 4 has been removed and Table 1 has been transferred to the supplementary data section. We have also added a new figure to the revised manuscript (Fig. 2, shown below).

Figure 2. Venn diagram showing the distribution of proteins identified in the qualitative proteomic analysis across each of the 4 study groups.

  • Section 2.2 containing the quantitative analysis for sure is more interesting, but for me it was impossible to understand why certain proteins were chosen and others not.

ANSWER: We agree with the reviewer that the process by which specific proteins were selected could be more clearly explained. When we performed a SWATH-MS analysis, we identified many proteins for which significant differences in expression were observed between groups. From these proteins, we selected those with a p-value<0.05 and a fold change (relative to another group) >1.5. This information is now made clear in the legends of Tables 1–4, and in the Materials and Methods section. Furthermore, we restricted our selection to proteins involved (based on information in the uniprot database)in bone growth, function or metabolism.

  • More specifically: Table 2 states that 2 proteins are downregulated in UG and 1 protein in ERT-a and –b, but Table 3 contains a long list of downregulated proteins.What are table 2 and 3 referring to? Why are the numbers so different? What does “2 only in ERT-a” mean in Table 2?

ANSWER: Table 1 in the revised manuscript (Table 2 in the original manuscript) lists the proteins for which we observed changes in plasma levels in any of the patient groups (UG,ERT-a, or ERT-b)relative to controls. This table thus summarizes all proteins for which changes were observed relative to the controls, and that fulfilled the following conditions: p <0.05 and fold change>1.5. For greater clarity, the column title “groups” has been changed to “group comparison”, and additional information has been added to the legend, as follows:

“Table 1.Changes in plasma protein expression detected by SWATH in any of the patient groups (UG, ERT-a, or ERT-b) relative to the control group.”

Table 2 in the revised manuscript (Table 3 in the original manuscript)lists the proteins for which we observed changes in plasma levels in all patient groups (UG,ERT-a, and ERT-b) relative to controls. These data provide information of proteins that may be dysregulated in MPS IVA. For greater clarity, the table legend has been edited as follows:

“Table 2. Changes in plasma protein expression detected in all patient groups (UG, ERT-a, and ERT-b) relative to the control group.”

The difference between the numbers of altered proteins included in each table is because Table 1 shows comparisons of individual patient groups with controls (and therefore contains fewer proteins),while Table 2 shows the comparison of all patient groups combined (UG, ERT-a and ERT-b) compared with controls.

The comparisons of individual patient groups with controls allows the identification of proteins that may be dysregulated as a consequence of the disease (untreated patients vs controls) as well as proteins that undergo changes in expression in response to treatment (ERT-a versus controls and ERT-b versus controls). By contrasting this information with that obtained following comparison of the 3 combined patient groups with the control group, we can identify proteins that may be implicated in thedisease but not fully corrected by ERT.

  • In the text, GPX3 is emphasized, but somehow did not make it into the group of biomarkers. Why not? GPX3 activity can be measured easily and might therefore be useful as a biomarker.
  • ANSWER: Based on the reviewer’s suggestion, we carried out a more exhaustive literature search forGPX3 and its possible role in bone. It is known that oxidative stress can lead to more adverse effects in lysosomal diseases. However, the information in the literature related to the role of this protein  about the high expression of GPX3 participates in oxidative stress processes generating osteoporotic processes, and  low levels  may be related to high levels of methylation as occurs in pathologies such as for example Kashin-Beck disease generating an increase in apoptosis of chondrocytes. However, the knowledge in the function of this protein requires more studies. In our population of MPS IVA patients, plasma levels of this protein were decreased compared with controls. Nonetheless, a role of this protein in MPS IVA should not be ruled out. The graph below provides additional insight into the changes in GPX3 expression observed in our study population.

Box plot depicting GPX3expression in the four study groups. Each data point represents the median protein expression value obtained in an individual sample. The line inside the box represents the median of all values obtained. The upper and lower limits of the box represent the first and third quartiles. Whiskers represent the minimum and maximum values within 1.5 times the interquartile range.

  • Figure 8 and 9: These figures show the “most relevant plasma proteins”, but why are they most relevant? According to Table 3, they neither have the highest fold change nor the highest significance value?

ANSWER: Many proteins showed significant changes in expression (p<0.05 and FC>1.5) in MPS IVA patients with respect to controls. However, our analysis was performed using plasma samples. It is therefore possible that many of the proteins for which significant changes were observed are not candidate bone biomarkers (which may be better identified by analyzing cartilage samples). Therefore, the “most relevant” proteins are those that have a P<0.05 and a FC> 1.5 and we considered most likely to be implicated in MPS IVA, despite not necessarily showing the greatest fold change or lowest p-value

  • Figure 11 and text on the page before state that in the untreated group 4 proteins are significantly different from all other groups. Wouldn’t those be the best biomarkers for treatment success? However, those 4 proteins are different from the 4 proteins that were in the end chosen as putative biomarkers: in Figure 11 these are LV321, KVD28, A1AT and S10A9, while in the end FETUA, VINT, A1AT and CLUS were postulated as biomarkers?

ANSWER: We selected these proteins (FETUA, VINT, A1AT and CLUS) based on their role in bone interactions that may be associated with MPS IVA. While certain proteins showed more significant changes in expression, these were not selected as candidate biomarkers due to the absence of any information in the literature suggesting a role in the pathology of MPS IVA.

  • How does Table 4 relate to Figure 11, because in Table 4 a lot more than 4 proteins are listed?

ANSWER: Figure 12 (Fig 11 in the original manuscript) shows all the proteins present in the comparison of the UG group against the rest of the groups, in which we only marked the most significant proteins, In table 3 (Table 4 in the original manuscript)we show only the significant proteins of said study.

  • I understand that A1AT might be a good candidate. However, this should be confirmed with another method.

ANSWER: This protein has been identified as a potential biomarker of MPS IVA using multiplexed quantitative assays:

Martell, L.; Lau, K.; Mei, M.; Burnett, V.; Decker, C.; Foehr, ED. Biomarker analysis of Morquio syndrome: identification of disease state and drug responsive markers. Orphanet J Rare Dis.2011; 6:84.

  • Table 5: What is the significance of this table? If this lists proteins that differ between ERT-a/b and all other groups, does this mean healthy probands and untreated patients show the same expression?

ANSWER: Table 4 (Table 5 in the original manuscript) lists the proteins for which changes in expression were detected in the ERT-a or ERT-b groups relative to all other groups (ERT-a or ERT-b, UG plus CG). This comparison allows identification of proteins for which changes in expression occur as a result of enzyme replacement therapy (ERT), as they differ from both CG and UG. It does not imply that CG and UG have the same protein expression values.

Reviewer 2 Report

The authors present a proteomic study on a small number of MPS IVA samples and 6 controls. The paper was very hard to wade through to decipher what are the relevant and/or significant findings of the study, it is too long, repetitive and needs restructuring. I have some suggestions itemised below that I believe would improve the readability and comprehension of the study. Unfortunately there were no page or line numbers on the document to reference.

  1. The Introduction can be shortened. There is no need to provide the background on proteomics – it is not new technology – but rather focus on the findings of previous proteomic studies that have been conducted for the MPS disorders, and importantly what has been found.
  2. The authors need to mention the use of a CS disaccharide as a biomarker for MPS IVA. The authors described GAG and KS but using older methodologies and do not discuss contemporary biomarkers for MPS IVA. Please include Chin et al JIMD Rep 2020 55:68-74 in the very least.
  3. Please provide the details of the samples. Using only 6 plasma controls were used. Are they age/sex matched to the IVA samples? Can the authors also please discuss the variation in these six samples – this is especially important given that there are many more variables tested than samples.
  4. Much of the introduction is repeated in the discussion and much of the discussion is in the results. Please subdivide into subheadings and maybe this could be done using the proteins identified and quantified (putting them together not separately) according to system eg haemostasis, bone, immune system etc.
  5. In table 1 what was the rationale for allowing one sample not to include the proteins identified? Was it the same sample? how many common proteins across the four groups? Within the groups of the same sample there is considerable disparity in the number of proteins identified eg UG 158-268 that is a different of 110 proteins – nearly as many as that identified, even in the small control cohort of six there is a difference 57? I am not sure that I understand the relevance of this table.
  6. Figs 2, 3, 5 and 6 I think should be removed. These are uninformative, there are no error bars it is not clear what proteins are included, how many and what differences there are. This is further complicated by the lack of detail in the results which states: “Once we determined the proteins characteristic of each group….”, leaves the question of which proteins the authors are referring to.
  7. I cannot read Fig 4 and not sure what it is meant to show.
  8. Maybe a heat map would be better to show the data – this could cover what is in tables 3-5 too.
  9. The enzyme activity is shown in the methods, have the authors attempted to correlate this with the protein biomarkers or any other parameter – KS storage, age of onset?
  10. There is no evidence that the proteins implicated in coagulation have any relation to Gal6S needing to stabilise. This should be removed and the results are not the correct place for interpretive comments.
  11. I am not sure why the authors have separated out the SWATH data – it has been used for quantitative proteomics in many previous studies – as, in my opinion, this is where the research report should start. Can the authors please explain how they decided the proteins plotted in Figs 8-10 were “most relevant”?
  12. Previous works have shown IL elevated in MPS IVA (see Fujitsuka et al MGM Rep 2019 19:100455) can the authors explain why their study did not?
  13. The authors need to acknowledge that this a preliminary study and tone down their claims. With such a small number of samples, no other disease or MPS type included, these data show a proteomics technique applied to MPS IVA that identified some alterations in a few proteins (that could be due to chance alone) and did not identify any potentially useful biomarkers.
  14. Lastly, the manuscript needs a careful proofread for grammar and typographical errors. Even the abstract has an “and” before the word healthy controls.

Author Response

REVIEWER 2.

  • The authors present a proteomic study on a small number of MPS IVA samples and 6 controls. The paper was very hard to wade through to decipher what are the relevant and/or significant findings of the study, it is too long, repetitive and needs restructuring. I have some suggestions itemised below that I believe would improve the readability and comprehension of the study. Unfortunately there were no page or line numbers on the document to reference.

ANSWER: We thank the reviewer for their comments. We agree that study objectives could be described more clearly. The two main objectives of this study of plasma proteins are:

1- To assess the validity of potential biomarkers of MPS IVA proposed by other authors.

2- To identify new candidate biomarkers of MPS IVA

We have edited the abstract and introduction to clarify these objectives.

  • The Introduction can be shortened. There is no need to provide the background on proteomics – it is not new technology – but rather focus on the findings of previous proteomic studies that have been conducted for the MPS disorders, and importantly what has been found.

ANSWER: We have simplified the description of proteomics in the introduction. Noother proteomics studies of MPSIVA patients have been published, only our previously published paper (corresponding references are included in the bibliography)..

  • The authors need to mention the use of a CS disaccharide as a biomarker for MPS IVA. The authors described GAG and KS but using older methodologies and do not discuss contemporary biomarkers for MPS IVA. Please include Chin et al JIMD Rep 2020 55:68-74 in the very least.

ANSWER: Based on the reviewer’s suggestion we have cited this study and included this biomarker in the discussion section. It should be noted that this candidate biomarker has only been studied in a small number of MPS IVA samples to date. 

  • Please provide the details of the samples. Using only 6 plasma controls were used. Are they age/sex matched to the IVA samples? Can the authors also please discuss the variation in these six samples – this is especially important given that there are many more variables tested than samples.

ANSWER: We agree that insufficient information on sample provenance was provided. This study is a continuation of a previously published study, in which the patient’s characteristics are provided in detail. Readers are directed to this reference for further information.

Patient

ID

Sex

Age at Diagnosis

(y)

ERT

Age at Start of Treatment

(y)

Current Characteristics

Age

(y)

Height

(cm)

6 Minute Walk Test (m)

FVC

(mL)

FEV1

(mL)

1

F

1

No

-

31

98

250

600

500

2

M

2

No

-

31

113

305

870

700

3

M

2

No

-

21

95

ND *

380

260

4

M

2

No

40

99

ND *

480

360

5

F

4

No

-

15

103

341

920

820

6

F

3

No

-

29

99

ND *

ND **

ND **

7

F

1

No

-

18

119

272

110

900

8

M

unknown

No

-

21

103

ND *

920

700

9

M

1

Yes

12

16

100

105

690

450

10

M

2

Yes

2

6

104

450

770

720

11

M

3

Yes

13

18

113.5

472

1390

1330

12

M

3

Yes

11

19

113

234

1350

1160

13

M

5

Yes

18

22

110

344

870

730

Álvarez, J.V.; Bravo, S.B.; Chantada-Vazquez, M.P.; Colón, C.; De Castro, M.J.; Morales, M.;Vitoria, I.; Tomatsu, S.; Otero-Espinar, F.J.; Couce, M.L. Characterization of New Proteomic Biomarker Candidates in Mucopolysaccharidosis Type IVA. Int J Mol Sci. 2020. 22(1):226.

The following information has also been included in the Results section:

 “Plasma samples were acquired from 13 MPS IVA patients and 6 controls and classified as follows: healthy controls; untreated MPS IVA patients; and MPS IVA patients who had been receiving weekly ERT infusions for a mean of 7 years. From these ERT-treated patients, samples were acquired 24 h before (ERT-a) and 24 h after (ERT-b) one of their weekly infusions (Fig. 1). The demographic features of participating MPS IVA patients have been previously described [33].”

  • Much of the introduction is repeated in the discussion and much of the discussion is in the results. Please subdivide into subheadings and maybe this could be done using the proteins identified and quantified (putting them together not separately) according to system eghaemostasis, bone, immune system etc.

ANSWER: Following the reviewer’s recommendation we classified all identified proteins into different sections. The results of the qualitative analysis are now included in the supplementary data, while the proteins identified in the quantitative analysis are presented in actual tables 2, 3 and 4 in which the most relevant candidate biomarkers are shown. We have also added a heatmap plotting the results of the SWATH quantitative analysis in which the relative abundance of each protein is color coded based on the z-score of the protein’s normalized peak area. The proteins are also classified according their function, as indicated by the color coding in the sidebar on the right of the heatmap.

Figure 8. Heatmap depicting the expression profiles of different proteins (also shown in Tables 2–4) as determined by SWATH quantitative analysis. The relative abundance of each protein is reflected by the colors in the heatmap, based on the z-score of the protein’s normalized peak area. The dendrogram represents the biological replicates of the UG, ERT-a, ERT-b, and CG groups. Proteins, listed in the sidebar on the right, are color coded according to their function, as follows: apolipoprotein-related proteins, dark green; blood coagulation, purple; calcium binding, dark orange; carrier proteins, light green; cytoskeletal proteins, light pink;  immune system, grey; metabolic processes, yellow; metabolic interconversion, light blue; other functions, white; protein binding, pink; transport, light grey.

  • In table 1 what was the rationale for allowing one sample not to include the proteins identified? Was it the same sample? how many common proteins across the four groups? Within the groups of the same sample there is considerable disparity in the number of proteins identified eg UG 158-268 that is a different of 110 proteins – nearly as many as that identified, even in the small control cohort of six there is a difference 57? I am not sure that I understand the relevance of this table.

ANSWER: Table 1 shows the number of proteins identified by DDA (qualitative analysis) in individual samples. However, in order to perform functional analysesthe mean expression level per group needs to be obtained. The qualitative analysis has some limitations (explained in the Study limitations section). In some cases a certain protein may not be detected in a given sample, due to that this proteins may be masked by other most abundant protein. To ensure adequate representation of proteins per group, we performed a restrictive selection including only proteins that were identified in n-1 samples per group. Thus, proteins that were identified in only 1 or 2 samples were not included in our functional analysis. While this approach may result in the exclusion in certain proteins of potential interest, it ensures that all proteins included in our functional analysis are adequately represented in each sample.

It should be noted that plasma samples were depleted to eliminate the most abundant proteins prior to the proteomic analysis. However, this process is not fully effective. This complicates the proteomic analysis and can result insubstantial differences in the number of proteins identified per sample.

In line with the suggestion of reviewer 1, Table 1 from the original manuscript has been moved to the supplementary data, and we have added a Venn diagram (Figure 2 in the revised manuscript) showing the distribution of proteins identified in the qualitative proteomic analysis across each of the 4 study groups.

  • Figs 2, 3, 5 and 6 I think should be removed. These are uninformative, there are no error bars it is not clear what proteins are included, how many and what differences there are. This is further complicated by the lack of detail in the results which states: “Once we determined the proteins characteristic of each group….”, leaves the question of which proteins the authors are referring to. I cannot read Fig 4 and not sure what it is meant to show.

ANSWER: We agree with the reviewer's comment regarding Figures 2,3,5, and 6. These figures were generated using FunRich (Functional Enrichment analysis tool. http://www.funrich.org/),aprogram designed in 2015to perform functional analyses of qualitative proteomic data.

Pathan M, Keerthikumar S, Ang CS, Gangoda L, Quek CY, Williamson NA, Mouradov D, Sieber OM, Simpson RJ, Salim A, Bacic A, Hill AF, Stroud DA, Ryan MT, Agbinya JI, Mariadason JM, Burgess AW, Mathivanan S. FunRich: An open access standalone functional enrichment and interaction network analysis tool. Proteomics. 2015;15(15):2597-601.

Benito-Martin A, Peinado H. FunRich proteomics software analysis, let the fun begin! Proteomics. 2015;15(15):2555-6.

This program is used extensively by the scientific community to perform functional analyses of qualitative proteomic data between samples, given the differences in the percentage protein expression found in UNIPROT databases. Moreover, the program can provide information about cellular behavior, molecular function, biological processes, and protein domains.

This type of analysis is not a statistical analysis, but rather detects differences in protein expression in the context of the function of interest. that the impossibility of performing a statistical analysis is a limitation of the program used for this functional analysis.

Moreover, the proteins depicted in these figures were selected based on alterations previously described in MPS IVA. In almost all cases, our findings were in line with that which is already known about the pathology (e.g. the increase in the levels of specific interleukins in MPS IVA patients).

In accordance with the reviewer’s suggestions, Figure 4 has been removed and Table 1 has been transferred to the supplementary data section. We have also added a new figure to the revised manuscript (Fig. 2, shown below).

Figure 2. Venn diagram showing the distribution of proteins identified in the qualitative proteomic analysis across each of the 4 study groups.

  • Maybe a heat map would be better to show the data – this could cover what is in tables 3-5 too.

ANSWER: In accordance with the reviewer’s suggestion, we have added a heatmap including color coding of proteins according to their function (Fig 8).

  • The enzyme activity is shown in the methods, have the authors attempted to correlate this with the protein biomarkers or any other parameter – KS storage, age of onset?

ANSWER: In this study we did not investigate correlations between protein biomarkers and other parameters, although this has been done in previous studies published by our group. For example, we have previously described a correlation between GALNS expression and keratan and chondrotin 6 sulfate deposits.

Álvarez, JV.; Bravo, SB.; García-Vence, M.; De Castro, MJ.; Luzardo, A.; Colón, C.; Tomatsu, S.; Otero-Espinar, F.J.; Couce, M.L. Proteomic Analysis in Morquio A Cells Treated with Immobilized Enzymatic Replacement Therapy on Nanostructured Lipid Systems. Int J Mol Sci. 2019; 20(18).

 Álvarez J V, Herrero Filgueira C, de la Fuente González A, Colón Mejeras C, Beiras Iglesias A, Tomatsu S, Blanco Méndez J, Luzardo Álvarez A, Couce ML; Otero Espinar FJ .Enzyme-Loaded Gel Core Nanostructured Lipid Carriers to Improve Treatment of Lysosomal Storage Diseases: Formulation and In Vitro Cellular Studies of Elosulfase Alfa-Loaded Systems.Pharmaceutics.2019, 11, 522.

  • There is no evidence that the proteins implicated in coagulation have any relation to Gal6S needing to stabilise. This should be removed and the results are not the correct place for interpretive comments.

ANSWER: We agree with the reviewer that no clear evidence has been published indicating a direct relationship between hemostatic phenomena and MPS IVA. We remove the text related to this subject according your suggestions.

I am not sure why the authors have separated out the SWATH data – it has been used for quantitative proteomics in many previous studies – as, in my opinion, this is where the research report should start. Can the authors please explain how they decided the proteins plotted in Figs 8-10 were “most relevant”?

ANSWER: We feel that the use of quantitative and qualitative analyses provides a more complete perspective on the pathology, and that the large amount of information that can be acquired with these approaches allows us to develop a greater body of  knowledge about the disease and to identify proteins that may serve as biomarkers of disease progression. Our group has previously published studies that demonstrate the validity of these approaches in the search for candidate biomarkers of MPS IVA.

  • Can the authors please explain how they decided the proteins plotted in Figs 8-10 were “most relevant”?

ANSWER: Many proteins showed significant changes in expression (p<0.05 and FC>1.5) in MPS IVA patients with respect to controls. However, our analysis was performed using plasma samples. It is therefore possible that many of the proteins for which significant changes were observed are not candidate bone biomarkers (which may be better identified by analyzing cartilage samples). Therefore, the “most relevant” proteins are those that have a P<0.05 and a FC> 1.5 and we considered most likely to be implicated in MPS IVA, despite not necessarily showing the greatest fold change or lowest p-value

  • Previous works have shown IL elevated in MPS IVA (see Fujitsuka et al MGM Rep 2019 19:100455) can the authors explain why their study did not?

ANSWER: In our qualitative analysis we searched for ILs that showed increases in expression, and examined how expression levels changed in response to ERT. In addition to increases in the expression of IL-1β in MPS IVA (UG), we also observed changes in other ILs that activate the inflammation pathway, including interferon γ, IL-6, and IL-8. We agree, there are different studies, as the paper mention by this reviewer, but differences between the 2 studies in terms of the proteomic techniques used and the parameters studies could account for these discrepancies.

The authors need to acknowledge that this a preliminary study and tone down their claims. With such a small number of samples, no other disease or MPS type included, these data show a proteomics technique applied to MPS IVA that identified some alterations in a few proteins (that could be due to chance alone) and did not identify any potentially useful biomarkers.

ANSWER: We agree that the number of samples is small. However, being a rare disease it is very difficult to obtain large numbers of samples to analyze. We are also aware that plasma samples are not the most optimal sample type in which to search for biomarkers, and that bone cartilage samples would be more suitable. This study sought to expand current knowledge on plasma proteins that may undergo changes in expression in MPS IVA, and thus could constitute potential diagnostic or prognostic biomarkers. We are aware of the limitations of the techniques used. We will continue to seek to identify the best possible sample type in which to search for disease biomarkers and to verify the validity of the proteins identified in the present study as biomarkers of MPS IVA.

  • Lastly, the manuscript needs a careful proofread for grammar and typographical errors. Even the abstract has an “and” before the word healthy controls.

ANSWER: We have carefully revised the manuscript and corrected any grammatical and typographical errors.

Round 2

Reviewer 1 Report

The manuscript was improved, most of my comments were addressed. However, no experiments to verify the identified biomarkers with a different method were performed.

Author Response

REVIEWER 1

The manuscript was improved, most of my comments were addressed. However, no experiments to verify the identified biomarkers with a different method were performed.

ANSWER: Thanks for the valuable suggestion, we agree that a validation of biomarkers using orthogonal techniques is necessary so that they can be used on a daily basis in clinical practice. However, it is necessary to bear in mind that this is an ultra-rare disease that appears in 1/250000 births, which makes it difficult to collect samples.

In other hand, the same samples used in this study were recently used in the characterization of new candidates for proteomic biomarkers in leukocytes from patients with MPS IVA. In this study we found some of the main candidate biomarker as VTNC. In addition, some of the biomarkers proposed in our study were studied in other articles using other techniques (Martell L.et al. Morquio syndrome biomarker analysis: identification of disease states and drug response markers. Orphanet J Rare Dis. 2011; 6: 84.)

But following the recommendation of the reviewer 1, to clarify this point, we have introduced this paragraph in the limitations of the study:

“The selected candidate proteins must be studied using techniques that are used daily in clinical practice to demonstrate their validity. However, the low number of patients with this disease makes this analysis difficult”.

Reviewer 2 Report

Thank you for addressing the concerns raised.

Author Response

REVIEWER 2

Thank you very much for your comment